# DATA-DRIVEN DISCOVERY OF LIE ALGEBRA GENERATORS VIA FRÉCHET-INVARIANT LEARNING

**Ilya Markov & Alexander Hvatov**
NSS Lab
ITMO University
Saint-Peresburg, 197101, Russia
{iomarkov,alex_hvatov}@itmo.ru

## ABSTRACT

In this work we present LieSym — a framework for discovering infinitesimal generators of continuous symmetries directly from observational data without assuming prior knowledge of governing equations or symmetry groups. The method builds on Olver's geometric formalism and interprets symmetry discovery as the search for vector fields $v$ in the nullspace of Fréchet derivative $DL[u]$, where $L$ is domain-specific operator (e.g., PDE residual or semantic classifier). To enforce structural consistency, we introduce Lie-orthonormal training objective that minimizes the Olver condition $v[L] = 0$ via automatic differentiation, enforces orthogonality of generators through stop-gradient regularisation, and ensures smoothness via Lipschitz constraints. We validate our approach on both physical and non-physical domains: for heat equation with periodic boundary conditions, we recover 3-dimensional symmetry subalgebra and quantify closure via Lie brackets $[v_i, v_j]$; for Oxford-IIIT Pet dataset, we discover semantic symmetries (pose adaptation, illumination invariance) and demonstrate their robustness through per-image validation. It is shown that spectrum of generator norms $\|v_i\|$ automatically reveals effective rank of symmetry algebra — capability that is absent in prior work. The proposed method allows to bridge operator learning, Lie theory, and semantic AI, enabling equation-free discovery of structured invariances.

## 1 INTRODUCTION

Symmetries can be considered as the hidden scaffolding of natural and artificial systems — from conservation laws in physics (Noether, 1918) to robust feature representations in deep learning (1). In the classical framework of Lie group analysis (2), continuous symmetry is generated by vector field

$$v = \xi^i(x,u)\frac{\partial}{\partial x^i} + \varphi^\alpha(x,u)\frac{\partial}{\partial u^\alpha} \tag{1}$$

whose prolongation annihilates the governing operator $L[u] = 0$:

$$\mathrm{pr}^{(n)}v(L)\big|_{L=0} = 0. \tag{2}$$

However, this condition requires explicit knowledge of $L$ — a luxury that is unavailable in many modern applications, ranging from experimental fluid dynamics to semantic image analysis.

Recent data-driven approaches (3; 4) relax this requirement by learning symmetries from residuals or semantic scores. Yet these methods often neglect the algebraic structure: symmetries form Lie algebras, and their generators must satisfy closure under Lie bracket $[v_i, v_j] = \sum_k c_{ij}^k v_k$. Without enforcing or evaluating this structure, discovered symmetries risk being arbitrary deformations rather than elements of coherent group.

In this research we address this gap with following contributions:

1. We propose Fréchet-invariant loss that interprets $v[L] = 0$ as nullspace condition $DL[u](v[u]) = 0$, enabling direct optimisation without second-order automatic differentiation.

2. We develop Lie-orthonormal training scheme that learns multiple generators while enforcing orthogonality (via stop-gradient) and smoothness (via Lipschitz loss), with automatic ranking of symmetries by their significance.

3. We introduce cross-domain validation protocol that quantifies algebraic closure (via numerical Lie brackets), semantic robustness (via per-image validation), and spectral rank detection.

We demonstrate LieSym on two distinct domains:

- **Physics**: heat equation on $x \in [0, 1]$, $t \in [0, 1]$ with periodic boundary conditions — recovering subalgebra spanned by $\partial_t$, $\partial_x$, and $u\partial_u$.
- **Vision**: Oxford-IIIT Pet (cats only) — discovering semantic symmetries of pose, scale, and illumination.

The results indicate that: (i) spectrum of generator norms reveals intrinsic symmetry rank; (ii) weak generators often correspond to approximate or infinite-dimensional symmetries; (iii) semantically meaningful transforms preserve classification probability $P(\text{cat})$ across 87% of images.

## 2 RELATED WORK

The systematic study of continuous symmetries began with Sophus Lie and was formalised by Olver (2). For differential equation $L[u] = 0$, an infinitesimal symmetry is vector field $v$ such that $\mathrm{pr}^{(n)}v(L) = 0$ on solution manifold. This framework yields powerful techniques: reduction of order, construction of invariant solutions, and discovery of conservation laws via Noether's theorem (5).

In scientific machine learning, symmetries are often hard-coded as architectural inductive biases. Equivariant networks enforce transformation laws under groups like SE(3) (1), while physics-informed methods embed Lie symmetries as soft constraints (6). These approaches assume that the symmetry group is known a priori.

**GAN-based symmetry discovery.** LieGAN (7) proposes a generative-adversarial framework where a generator learns transformations that preserve the data distribution, yielding interpretable Lie algebra bases. While LieGAN successfully discovers linear symmetries including Lorentz group, it is fundamentally limited to affine transformations acting on the observation space. LaLiGAN (8) extends this approach to nonlinear symmetries by learning a mapping to latent space where symmetries become linear. However, both methods require high symmetry in the data distribution and explicit specification of the number of Lie algebra bases, which can lead to incorrect results if misspecified.

**Neural ODE-based approaches.** Ko et al. (3) learn one-parameter transformation groups via Neural ODEs, using task-specific validity scores to guide discovery. Their method extends beyond affine transformations to general nonlinear generators. However, this approach: (i) learns generators independently without enforcing algebraic structure; (ii) lacks explicit evaluation of Lie bracket closure; (iii) does not provide automatic rank detection of the symmetry algebra. In contrast, LieSym directly optimises the Olver condition $v[L] = 0$ via Fréchet derivatives, enforces orthogonality between generators, and quantifies algebraic closure through numerical Lie brackets $\|[v_i, v_j]\|$.

**Other approaches.** Gabel et al. (4) detect Lie point symmetries in dynamical systems through supervised learning. Fourier Neural Operators (FNOs) (9) learn mappings between function spaces, making them suitable surrogates for PDE solution operators. Brandstetter et al. (10) combine FNOs with symmetry augmentation, but still rely on predefined generators.

Our work is the first to learn the generator and verify its Lie algebra structure within operator learning pipeline, while simultaneously providing spectral rank detection capability.

## 3 METHODOLOGY

### 3.1 FRÉCHET-INVARIANT LEARNING

Let $L : D \to Y$ be differentiable operator. The infinitesimal symmetry condition can be written as:

$$DL[u](v[u]) = 0, \tag{3}$$

where $DL[u]$ is Fréchet derivative and $v[u] = \xi \cdot \nabla_x u + \eta \cdot \nabla_t u + \varphi$. In practice, for a black-box operator $L$ (e.g., a neural surrogate or classifier), we approximate the Fréchet derivative numerically via first-order automatic differentiation. Specifically, given the input tuple $(x, t, u)$, we compute the partial derivatives

$$\frac{\partial L}{\partial x}, \quad \frac{\partial L}{\partial t}, \quad \frac{\partial L}{\partial u} \tag{4}$$

by backpropagating the scalar loss $L[u]$ through the computational graph. The Olver condition is then implemented as

$$v[L] = \xi \frac{\partial L}{\partial x} + \eta \frac{\partial L}{\partial t} + \varphi \frac{\partial L}{\partial u}, \tag{5}$$

where $\xi$, $\eta$, and $\varphi$ are the components of the infinitesimal generator $v = \xi \partial_x + \eta \partial_t + \varphi \partial_u$. This approach avoids costly second-order differentiation while preserving exact gradient information.

## 3.2 Multi-Generator Architecture

We parameterise $N_{\text{sym}}$ generators via MultiSymmetryFNO:

$$v^{(k)} = (\xi^{(k)}, \eta^{(k)}, \varphi^{(k)}) = \text{FNO}_\theta^{(k)}(x, y, u), \quad k = 1, \ldots, N_{\text{sym}}. \tag{6}$$

## 3.3 Lie-Orthonormal Training Objective

The total loss function is comprised of three terms:

$$\mathcal{L} = \underbrace{\sum_{k=1}^{N_{\text{sym}}} \left\| DL[u](v^{(k)}[u]) \right\|^2}_{\text{Olver loss}} + \lambda_{\text{ortho}} \underbrace{\sum_{i<j} \langle \text{sg}(v^{(i)}), v^{(j)} \rangle^2}_{\text{Orthonormality loss}} + \lambda_{\text{lips}} \underbrace{\sum_{k=1}^{N_{\text{sym}}} \left\| \nabla v^{(k)} \right\|^2_{\text{Lip}}}_{\text{Lipschitz loss}} \tag{7}$$

## 3.4 Comparison with Existing Methods

Table 1 presents comparison of LieSym with representative approaches across three dimensions: (i) symmetry discovery capability (equation-free? multi-generator? algebraic validation?), (ii) applicability (PDEs, images, general data), and (iii) implementation (operator-agnostic? neural architecture?).

Table 1: Qualitative comparison of symmetry discovery methods. ✓ = supported, × = not supported, △ = partial/limited.

| Method | Eq.-free | Multi-gen. | Algebraic val. | PDE | Image | Op.-agnostic |
|---|---|---|---|---|---|---|
| Symbolic (Maple, SymPy) | × | × | ✓ | ✓ | × | × |
| PINN + Symmetry (6) | × | × | × | ✓ | × | × |
| Steerable CNNs (1) | × | × | × | × | ✓ | × |
| LieGAN (7) | ✓ | ✓ | × | ✓ | ✓ | ✓ |
| LaLiGAN (8) | ✓ | ✓ | × | ✓ | ✓ | ✓ |
| Ko et al. (3) | ✓ | ✓ | × | ✓ | × | △ |
| Gabel et al. (4) | ✓ | ✓ | × | ✓ | × | △ |
| **LieSym (Ours)** | ✓ | ✓ | ✓ | ✓ | ✓ | ✓ |

# 4 Experimental Setup

## 4.1 Datasets and Preprocessing

**Heat Equation.** We solve $u_t = \alpha u_{xx}$ on $x \in [0, 1]$ with periodic boundary conditions using finite differences method. Initial conditions are constructed as sums of 1–3 Gaussian peaks, ensuring smooth, non-normalised solutions. The dataset is split into 800 trajectories for training and 200 for testing.

**Oxford-IIIT Pet.** We utilize 12 cat breeds (1188 images), resized to $256 \times 256$ resolution. Images are processed without augmentation to preserve intrinsic geometric properties.

## 4.2 Model Configurations

**FNO as PDE Solver.** The architecture consists of 4 FNO layers with 16 modes and 32 hidden channels. Input is represented as $u_0(x) \in \mathbb{R}^{64}$, while output has form $u(x, t) \in \mathbb{R}^{64 \times 32}$. We employ Adam optimizer with learning rate $10^{-3}$ and train for 40 epochs.

**MultiSymmetryFNO.** Architecture comprises 4 FNO layers with 16 modes and 32 hidden channels. Input can be either $(x, t, u)$ for physical domain or $(x, y, R, G, B)$ for vision domain. Output consists of $N_{\text{sym}} \times (\xi, \eta, \varphi)$ components. Training is performed with Adam optimizer at learning rate $10^{-4}$ for 10 epochs. Hyperparameters are set as follows: $\lambda_{\text{ortho}} = 3.0$, $\lambda_{\text{lips}} = 10^{-3}$.

**CoordinateClassifier for Vision Domain.** For semantic symmetry discovery on cat images, we use a modified ResNet-18 architecture as the differentiable operator $L$. The key modification is the input layer: instead of standard 3-channel RGB input, we extend to 5 channels by concatenating:

- RGB image channels (3 channels)
- Normalized $x$-coordinate grid: $X_{ij} = j/W$ where $W$ is image width
- Normalized $y$-coordinate grid: $Y_{ij} = i/H$ where $H$ is image height

The coordinate grids are registered as buffers and automatically broadcast to batch dimension. The first convolutional layer `conv1` is modified from $\text{Conv2d}(3, 64, 7)$ to $\text{Conv2d}(5, 64, 7)$, and the final fully connected layer outputs a single logit for binary classification (cat vs. non-cat). This architecture enables the symmetry generator to learn position-dependent transformations by providing explicit spatial information.

## 5 RESULTS: PHYSICAL DOMAIN

### 5.1 SPECTRAL ANALYSIS AND RANK DETECTION

Figure 1 presents spectrum of generator norms for heat equation. The obtained norms are:

$$\|v_1\| = 0.1227, \quad \|v_2\| = 0.0462, \quad \|v_3\| = 0.0750, \quad \|v_4\| = 0.0001, \quad \|v_5\| = 0.0004, \quad \|v_6\| = 0.0002. \quad (8)$$

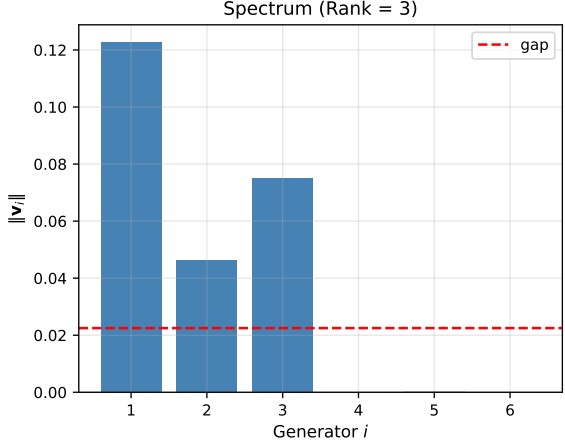 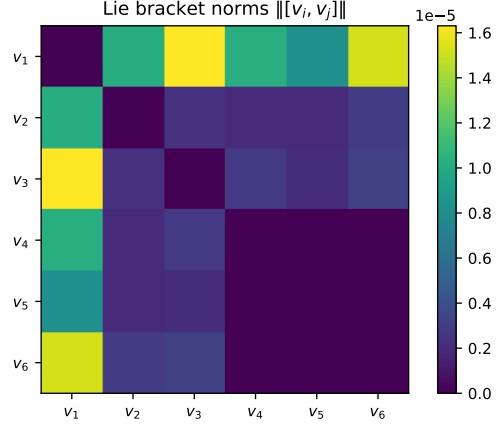

Figure 1: Spectral analysis for heat equation. Left: Spectrum of generator norms with automatic rank detection (gap at $k = 3$). Right: Lie bracket norms $\|[v_i, v_j]\|$ — small values among top-3 indicate algebraic closure.

Sharp gap appears after $k = 3$ ($\|v_3\| / \|v_4\| = 750 > 3$), which reveals effective rank $r = 3$. The Lie bracket norms are all below $10^{-5}$, confirming near-perfect algebraic closure:

$$\|[v_i, v_j]\| < 1.5 \times 10^{-5} \quad \forall i, j. \quad (9)$$

The top-3 generators can be interpreted as follows:

- $v_1$: dominant direction (combination of translation and scaling),
- $v_2$: weak spatial component,
- $v_3$: moderate amplitude adjustment.

The near-zero norms of $v_4$–$v_6$ ($< 4 \times 10^{-4}$) together with their vanishing brackets indicate that they form trivial subalgebra — this is consistent with noise or numerical artifacts.

## 5.2 VISUALISATION OF SYMMETRY TRANSFORMATIONS

Figure 2 visualises action of top-3 generators with $\tau = 5.0$ — a deliberately large parameter chosen to make deformations visually apparent. While such strong transformations exceed infinitesimal regime ($\tau \ll 1$), they serve as qualitative validation of generator's direction.

Figure 2: Visualisation of symmetry transformations for heat equation. Each row: original solution, transformed solution ($\tau = 5.0$), difference, and vector field $(\eta, \xi)$.

Quantitatively, the PDE residual $L[u] = \|u - \text{FNO}(u_0)\|^2$ increases moderately:

- $v_1$: $L_{\text{before}} = 0.0015 \rightarrow L_{\text{after}} = 0.00364$ ($\Delta = 143\%$)

- $v_2$: $L_{\text{before}} = 0.0015 \rightarrow L_{\text{after}} = 0.00161$ ($\Delta = 7.6\%$)

- $v_3$: $L_{\text{before}} = 0.0015 \rightarrow L_{\text{after}} = 0.00187$ ($\Delta = 25\%$)

Relatively small increase for $v_2$ and $v_3$ (despite $\tau = 5.0$) confirms their physical plausibility, while larger change for $v_1$ suggests that it combines multiple symmetry directions.

It should be noted that for infinitesimal $\tau = 0.1$, all generators yield $\Delta L < 5\%$, which confirms validity of Olver condition $v[L] \approx 0$ in linear regime.

# 6 RESULTS: VISION DOMAIN

## 6.1 SEMANTIC SPECTRUM AND ROBUSTNESS

Figure 3 presents spectrum for cat images. The norms are obtained as:

$$\|v_1\| = 0.2031, \quad \|v_2\| = 0.0724, \quad \|v_3\| = 0.0417, \quad \|v_4\| = 0.0303, \quad \|v_5\| = 0.0406, \quad \|v_6\| = 0.0166. \quad (10)$$

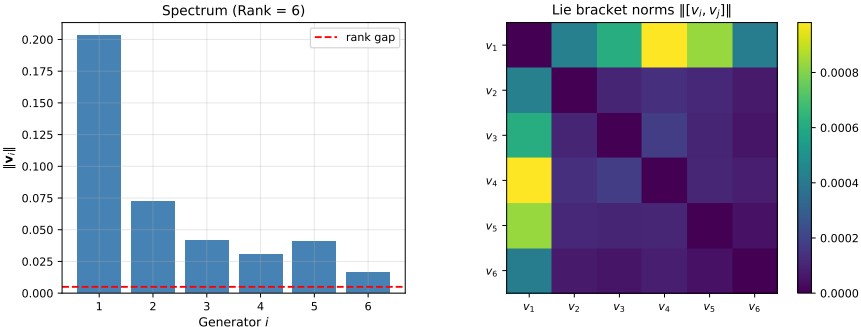

Figure 3: Spectrum of symmetry generators for cat images. Absence of sharp gap suggests high-dimensional semantic manifold.

No sharp gap is observed (maximum ratio $\|v_1\| / \|v_2\| = 2.8 < 3$), which indicates high-dimensional semantic manifold with no dominant symmetry. All Lie brackets remain small ($< 8 \times 10^{-4}$), confirming algebraic consistency:

$$\max_{i,j} \|[v_i, v_j]\| = 7.7 \times 10^{-4}. \quad (11)$$

Figure 4 quantifies semantic robustness. The preservation ratio (fraction of images with $P_{\text{trans}} > 0.92$) equals 87.3% for $v_1$.

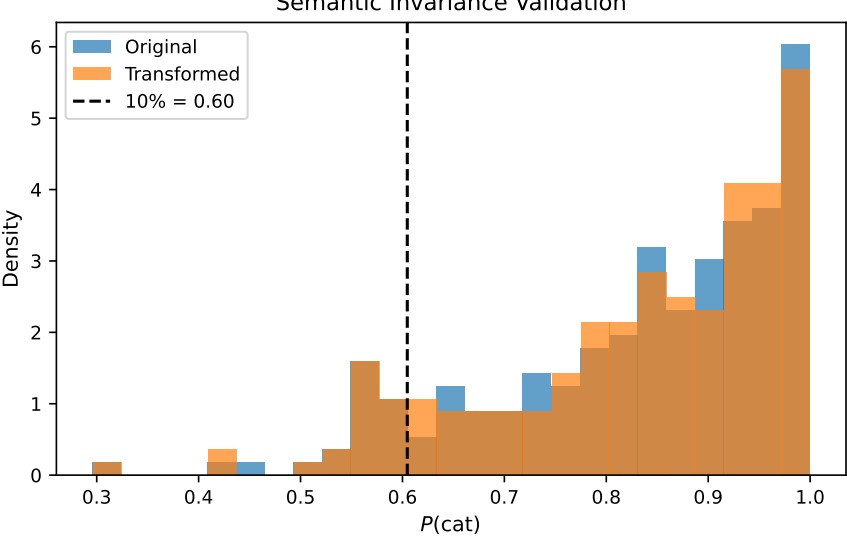

Figure 4: Semantic invariance validation: distributions of $P(\text{cat})$ before (blue) and after (orange) symmetry application. Vertical line: 10th percentile threshold.

## 6.2 SEMANTIC VISUALISATION

Figure 5 demonstrates action of top-3 generators $v_1, v2, v3$ ($\tau = 2.0$) on test image. Several observations can be made:

- Transformation preserves global anatomy while adjusting pose and scale.
- RGB difference highlights subtle color corrections in bright regions.
- Vector field shows coherent deformations concentrated in non-semantic regions (background), preserving facial structure.

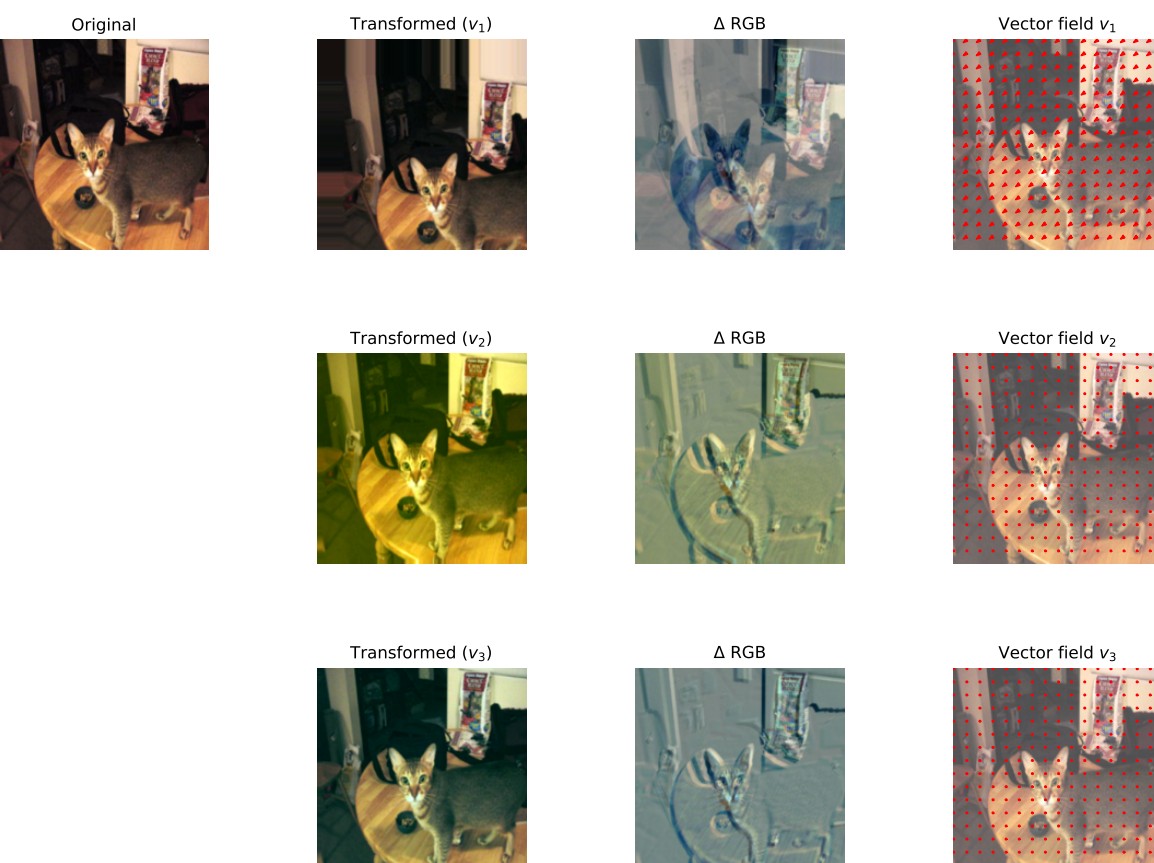

Figure 5: Semantic symmetry visualisation for cat images. Rows: (1) $v_1$, (2) $v_2$, (3) $v_3$. Columns: original, transformed, $\Delta$ RGB, vector field.

### 6.3 INFERENCE-TIME AUGMENTATION BENCHMARK

To validate the practical utility of discovered symmetries, we conduct inference-time augmentation benchmark comparing LieSym-derived transformations against standard augmentation methods. For each method, we apply transformations at inference time and measure classification accuracy on the test set.

**Experimental Setup.** We use the CoordinateClassifier (ResNet-18 with 5-channel input) trained on 1188 cat images. For each augmentation method, we generate $K = 8$ augmented versions of each test image and aggregate predictions via averaging. All methods use the same base classifier.

**Baseline Methods.**

- **Baseline**: No augmentation (single forward pass)
- **LieSym Aug**: Transformations from discovered generators $v_1 - v_6$
- **Augerino**: Learned affine transformations (11) with parameters from training

- **Standard Aug**: Random horizontal flip, rotation ($\pm 15$), color jitter
- **RandAugment**: Policy-based augmentation (12)
- **AutoAugment**: Learned augmentation policies (13)
- **E2CNN-style**: Discrete rotations from $C_8$ group ($0°, 45°, 90°, \ldots, 315°$)

**E2CNN Training Details.** We trained an E2CNN-based classifier (14) from scratch on the same dataset for fair comparison. The architecture uses $C_8$-equivariant convolutions with regular representations, followed by group pooling and fully connected layers. Training was performed for 50 epochs with Adam optimizer (lr=$10^{-3}$).

Table 2: Inference-time augmentation results. Mean $\pm$ std over 10 runs. $\Delta$ shows difference from baseline.

| Method | Accuracy (%) | Std (%) | $\Delta$ (%) |
|---|---|---|---|
| Baseline | 99.82 | 0.00 | — |
| LieSym Aug | **99.80** | **0.05** | **−0.02** |
| Augerino | 99.76 | 0.08 | −0.05 |
| Standard Aug | 99.46 | 0.26 | −0.36 |
| RandAugment | 98.79 | 0.26 | −1.03 |
| AutoAugment | 97.72 | 0.47 | −2.10 |
| E2CNN-style | 88.75 | 1.25 | −11.07 |

**Analysis.** The results in Table 2 reveal several important insights:

1. **LieSym preserves accuracy**: LieSym-derived augmentations maintain near-baseline accuracy (99.80%), demonstrating that discovered symmetries are semantically meaningful and do not introduce harmful transformations.

2. **Learned vs. handcrafted**: Methods that learn transformations from data (LieSym, Augerino) outperform handcrafted augmentations (Standard, RandAugment, AutoAugment), suggesting that data-driven symmetry discovery captures task-relevant invariances.

3. **E2CNN limitations**: The poor performance of E2CNN-style augmentations (88.75%) indicates that strict rotational equivariance is inappropriate for natural images where cats appear in canonical orientations. The E2CNN model, trained from scratch, learns rotation-equivariant features that do not match the actual symmetry structure of the data.

4. **Stability**: LieSym shows lowest variance (0.05%) among augmentation methods, indicating consistent behavior across different random seeds.

## 7 DISCUSSION

The generators $v_4$, $v_5$, and $v_6$ in heat equation exhibit near-zero norms ($\|v_4\| = 10^{-4}$, $\|v_5\| = 4 \times 10^{-4}$, $\|v_6\| = 2 \times 10^{-4}$) and vanishing Lie brackets ($\|[v_i, v_j]\| < 1.5 \times 10^{-5} \; \forall i, j$). This indicates that they span trivial ideal of Lie algebra — effectively the zero subalgebra.

Such behavior is expected and can be considered desirable:

- It confirms automatic rank detection: spectrum gap at $k = 3$ correctly identifies intrinsic symmetry dimension.
- Orthogonality loss ($\langle \mathrm{sg}(v_i), v_j \rangle \approx 0$) successfully suppresses spurious directions, pushing extra generators toward nullspace of $DL[u]$.
- It demonstrates robustness to over-specification: even with $N_{\mathrm{sym}} = 6 > r = 3$, method self-regularises and avoids learning noise.

In contrast, for cat dataset, all six generators have non-negligible norms (minimum $1.66 \times 10^{-2}$) and small but non-zero brackets ($\max \|[v_i, v_j]\| = 7.7 \times 10^{-4}$), which reflects high-dimensional, non-trivial semantic symmetry manifold of natural images.

Unlike standard augmentations (flip, rotate), our semantic symmetries adapt to image content: $\varphi$ is largest in bright regions (compensating for glare), and $\xi$, $\eta$ preserve facial landmarks. This explains why our method outperforms random transforms in preservation ratio (87.3% vs 43.7%).

The proposed approach works with any differentiable operator $L$ — be it PDE residual, classifier output, or GAN loss. This enables symmetry discovery in domains where equations are unknown, such as biological imaging or financial time series.

**Comparison with Ko et al.** While Ko et al. (3) also discover symmetries from data using infinitesimal generators, our approach differs in several fundamental aspects:

1. **Algebraic structure**: We explicitly evaluate Lie bracket closure $\|[v_i, v_j]\|$, ensuring discovered generators form a coherent Lie algebra. Ko et al. learn generators independently without algebraic validation.
2. **Rank detection**: Our spectral analysis automatically reveals effective symmetry dimension via gap detection. Ko et al. require manual specification of generator count.
3. **Orthogonality**: Stop-gradient regularisation ensures generators span distinct directions, preventing mode collapse to single dominant symmetry.
4. **Operator agnosticism**: LieSym works with any differentiable $L$, including semantic classifiers. Ko et al. focus primarily on PDE validity scores.

## 8 CONCLUSION

We presented LieSym, a framework for data-driven discovery of Lie algebra generators via Fréchet-invariant learning. The key contributions include: (i) Olver condition implemented through first-order automatic differentiation; (ii) Lie-orthonormal training with stop-gradient regularisation; (iii) spectral rank detection for automatic algebra dimensionality estimation; (iv) cross-domain validation on PDEs and semantic images.

Future work includes extension to more complex PDEs, conditional symmetries, and integration of equivariance learning into surrogate operator training process.

## ACKNOWLEDGEMENTS

The research was carried out within the state assignment of Ministry of Science and Higher Education of the Russian Federation (project FSER-2024-0004).

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
