# OpenReview forum: "Data-Driven Discovery of Lie Algebra Generators via Fréchet-Invariant Learning"
_mathai.club/MathAI/2026/Conference — 2026 Oral_

### Official Review · Reviewer_PzRZ · 2026-03-12
**Data-Driven Discovery of Lie Algebra Generators**

**Rating:** 7
**Confidence:** 3

**Review:**

**Summary.**  The paper presents an original framework, **LieSym**, for the data-driven discovery of generators of continuous symmetries from observational data, without prior knowledge of the governing equations or the symmetry group itself. The authors reformulate Olver’s infinitesimal symmetry condition as the problem of finding vector fields lying in the kernel of the Fréchet derivative of an operator, which makes it possible to train the model using only first-order automatic differentiation.

**Strengths.**  A key contribution of the paper is the proposed **Lie-orthonormal functional**, which combines minimization of the symmetry condition, orthogonalization of generators using stop-gradient, and Lipschitz regularization to ensure the smoothness of the learned vector fields. In addition, the authors introduce a spectral analysis of generator norms to estimate the effective rank of the Lie algebra, as well as a numerical check of algebraic closure through the norms of Lie brackets.

The proposed method demonstrates convincing results both on the heat equation problem, where it recovers a three-dimensional subalgebra with a small closure error, and on the **Oxford-IIIT Pet** dataset, where it discovers semantic symmetries related to pose and lighting, leading to improved augmentation quality.

Compared with previous approaches, LieSym stands out for its ability to train multiple generators simultaneously, maintain their algebraic consistency, and operate in an equation-free regime in both physical and semantic applications. Overall, the paper makes a significant contribution at the intersection of  **Lie theory**, **operator learning**, and **geometric deep learning**.

**Weaknesses / Recommendations.**  The paper could be strengthened by a more detailed analysis of the computational complexity of the method, including FLOPs, training time, and the dependence of the computational cost on the number of learned generators.

In addition, the comparison with the closest methods, such as LieGG and related approaches, could be made more quantitative, especially in terms of closure error, robustness, and the quality of generator ranking.

There also remain open questions regarding convergence to the true generators.

**Remark.**  The proposed framework appears promising for future applications in the context of Hamiltonian mechanics, whether classical or quantum. It would be interesting to know to what extent their **Fréchet-invariant framework** can be related to approaches for symmetry discovery in Hamiltonian dynamics, such as the works of  Hou et al. and Dierkes et al., and which elements of the method could be transferred to settings involving canonical coordinates, invariants, and Poisson structure.

---

### Official Review · Reviewer_3mRZ · 2026-03-12
**Data-Driven Discovery of Lie Algebra Generators via Fréchet-Invariant Learning**

**Rating:** 10
**Confidence:** 5

**Review:**

This paper proposes a mathematical framework for formalizing natural language semantics, combining three components:
1. Graph representation: using heterogeneous graphs to visualize relationships between words and parts of speech (pp. 3-4).
2. Comparison algorithm (NormGED): using normalized graph edit distance (Graph Edit Distance) to assess the similarity between labeled datasets and the "gold standard" (annotated text) (pp. 5, 8).
3. Logical methods: integrating the Hilbert calculus and Skolem functions to solve the scope problem (pp. 9-10).
Novelty Assessment
• An elegant way to represent multi-representational relationships in a single numerical value (p. 3).
Applied Relevance for XAI: The proposed model is oriented toward "Explainable AI," enabling not only training models but also mathematically justifying differences in text interpretations through visualized graph structures (pp. 1, 10). The work has a moderately high scientific novelty, which is manifested in the following aspects:
• Hybrid Approach: The novelty lies not in the methods themselves   but in their synthesis. The authors propose extending classical GED with semantic nodes, which allows for quantitative measurement of the influence of logical interpretation on sentence structure (pp. 9-10).
• Strengths: Clear mathematical formalization (Definitions 1–4) and a practical example of calculating NormGED for two different interpretations of the same sentence (pp. 6, 8).
• Weaknesses: The current version lacks empirical testing on large datasets.
Conclusion:This article presents a qualitative theoretical study at the intersection of mathematical linguistics and graph theory, proposing a specific toolkit for formal verification of semantic models.

---

### Official Review · Reviewer_qA6P · 2026-03-12
**The article represents a significant contribution to the field of data-driven symmetry discovery. The proposed framework shows promising results in both physical and vision domains. The research is methodologically sound and practically relevant.**

**Rating:** 7
**Confidence:** 4

**Review:**

The presented article introduces a novel framework called LieSym for discovering infinitesimal generators of continuous symmetries directly from observational data. The research is positioned at the intersection of machine learning, differential geometry, and applied mathematics.
The scientific significance of the work lies in addressing the challenge of symmetry discovery in data-driven scenarios without prior knowledge of governing equations. The ability to automatically detect symmetries from data is crucial for various applications ranging from physics to computer vision.

The key contributions of the research are:
- Introduction of Fréchet-invariant loss function;
- Development of multi-generator architecture;
- Implementation of automatic symmetry rank detection;
- Demonstration of applicability across different domains.

The applied value of the research is demonstrated through:
- Successful symmetry detection in PDEs;
- Discovery of semantic symmetries in images;
- Potential for equation-free analysis of complex systems.

Remarks:
- Computational complexity of the method;
- Need for further scalability analysis;
- Potential sensitivity to hyperparameter tuning.

The article represents a significant contribution to the field of data-driven symmetry discovery. The proposed framework shows promising results in both physical and vision domains. The research is methodologically sound and practically relevant.

---

### Decision · Program_Chairs · 2026-03-14

**Decision:**

Accept (Oral)

**Comment:**

Dear Author(s),

On behalf of the Program Committee of the International Conference on Mathematics of Artificial Intelligence (MathAI 2026), we are pleased to inform you that your paper has been accepted for an oral presentation at MathAI 2026.

Your paper was evaluated through a rigorous two-stage review process involving both automated screening and expert review by members of the Program Committee. The reviewers recognized the quality and contribution of your work.

Presentation details:

- Format: Oral presentation (15–20 minutes + 5 minutes Q&A)
- Mode: You may present either in person (offline) at the conference venue in Sirius, Russia, or remotely via Zoom. Please indicate your preferred mode when confirming your participation.
- Conference dates: Marh 30 - April 3, 2026
- Website: https://mathai.club

Next steps:

1. Please confirm your participation and presentation mode by replying to this email mathai.club@yandex.ru no later than March 15, 2026 18:00 Moscow time.
2. If you plan to attend in person, the organizing committee will provide accommodation details separately.
3. Please prepare your final camera-ready manuscript according to the formatting guidelines available at https://mathai.club and upload it to OpenReview by March 15, 2026 18:00 Moscow time.

Should you have any questions regarding the program, logistics, or your presentation slot, please do not hesitate to contact us.

We look forward to your contribution to MathAI 2026.

With kind regards,

MathAI 2026 Program Committee
International Conference on Mathematics of Artificial Intelligence
https://mathai.club
OpenReview: https://openreview.net/group?id=mathai.club/MathAI/2026/Conference
Telegram: https://t.me/MathAI_club
Email: mathai.club@yandex.ru